# Cognitive Improvement Effects of Electroacupuncture Combined with Computer-Based Cognitive Rehabilitation in Patients with Mild Cognitive Impairment: A Randomized Controlled Trial

**DOI:** 10.3390/brainsci10120984

**Published:** 2020-12-14

**Authors:** Jae-Hong Kim, Jae-Young Han, Gwang-Cheon Park, Jeong-Soon Lee

**Affiliations:** 1Department of Acupuncture and Moxibustion Medicine, College of Korean Medicine, DongShin University, Naju City 58245, Korea; nahonga@hanmail.net; 2Clinical Research Center, DongShin University Gwangju Korean Medicine Hospital, 141, Wolsan-ro, Nam-gu, Gwangju City 61619, Korea; smailcc@nate.com; 3Department of Physical and Rehabilitation Medicine, Chonnam National University Medical School and Hospital, Gwangju City 61469, Korea; 4Department of Nursing, Christian College of Nursing, Gwangju City 61662, Korea; mishilee@ccn.ac.kr

**Keywords:** mild cognitive impairment, electroacupuncture, computer-based cognitive rehabilitation, randomized controlled trial

## Abstract

This outcome assessor-blinded, randomized controlled clinical trial investigated the effects of electroacupuncture combined with computer-based cognitive rehabilitation (EA-CCR) on mild cognitive impairment (MCI). A per-protocol analysis was employed to compare the efficacy of EA-CCR to that of computer-based cognitive rehabilitation (CCR). Thirty-two patients with MCI completed the trial (EA-CCR group, 16; CCR group, 16). Patients received EA-CCR or CCR treatment once daily three days per week for eight weeks. Outcome (primary, ADAS-K-cog; secondary, MoCA-K, CES-D, K-ADL, K-IADL, and EQ-5D-5L) measurements were performed at baseline (week 0), at the end of the intervention (week 8), and at 12 weeks after completion of the intervention (week 20). Both groups showed significant changes in ADAS-K-cog score (EA-CCR, *p* < 0.001; CCR, *p* < 0.001) and MoCA-K (EA-CCR, *p* < 0.001; CCR, *p* < 0.001). Only the EA-CCR group had a significant change in CES-D (*p* = 0.024). No significant differences in outcomes and in the results of a subanalysis based on age were noted between the groups. These results indicate that EA-CCR and CCR have beneficial effects on improving cognitive function in patients with MCI. However, electroacupuncture in EA-CCR showed no positive add-on effects on improving cognitive function, depression, activities of daily living, and quality of life in patients with MCI.

## 1. Introduction

Mild cognitive impairment (MCI) is a condition in which individuals demonstrate a slight objective impairment in cognition (typically memory) that does not require help with the performance of activities of daily living [1,2,3]. MCI is considered an intermediate stage between the expected cognitive decline of normal aging and Alzheimer’s disease (AD), with a conversion rate of 5–10% per year [4,5,6,7]. Thus, MCI is a target for the prevention of AD development [8]. 

No high-quality evidence exists to support pharmacologic treatments for MCI [9]. Systematic reviews and meta-analyses evaluating the efficacy of cholinesterase inhibitors for MCI treatment have concluded that there is no convincing evidence that cholinesterase inhibitors have an effect on cognitive test scores or the progression of MCI to AD [10,11]. Some non-pharmacologic interventions, such as computerized cognitive training [12,13], exercise training [14], aerobic dance routine [15], and acupuncture [16,17], may be beneficial for patients with MCI. However, currently, no treatment method for MCI has been established [18].

Acupuncture is a common traditional Chinese medicine technique that is used for the treatment of various kinds of neurological disorders, including MCI [16]. Electroacupuncture (EA) treatment refers to the insertion of more than two needles into the skin and applying weak electricity through the needle [17]. EA has been reported to produce greater effect on neuroblast plasticity in the dentate gyrus [19], more widespread signal increases in the human brain as measured by functional magnetic resonance imaging [20] than acupuncture alone. A systematic review and clinical trials suggest that EA may have a beneficial effect on MCI [17,21,22,23,24]. Moreover, cognitive training could enhance brain activation in the areas related to memory [25]. Computer-based cognitive rehabilitation (CCR) has generated considerable attention as a safe, relatively inexpensive, and scalable intervention that aims to reduce cognitive decline in older adults [13]. Systematic reviews have demonstrated that CCR may generate some positive effects on patients with MCI or dementia [12,13].

Although evidence suggests that both EA and CCR have benefits on cognitive functions, evidence regarding the efficacy and safety of EA combined with CCR (EA-CCR) for treating MCI is insufficient. Hence, this study was performed to investigate the efficacy and safety of EA-CCR for the treatment of MCI and to determine whether EA has add-on effects by comparing EA-CCR with CCR alone in patients with MCI. 

## 2. Materials and Methods

This study followed the Standard Protocol Items: Recommendations for Interventional Trials (SPIRIT) and Consolidated Standards of Reporting Trials (CONSORT) statements (Appendix A). Details of the methods in this study have been reported previously [26].

### 2.1. Study Design

This study was a prospective, outcome assessor-blinded, single-center (DongShin University Gwangju Korean Medicine Hospital, Republic of Korea), randomized controlled trial with a 1:1 allocation ratio. A total of 36 participants who met the inclusion and exclusion criteria were randomly allocated to either the EA-CCR or the CCR group (*n* = 18 each group). Participants in the CCR group received RehaCom cognitive rehabilitation only, while those in the EA-CCR group had EA at the following acupoints: Baihui (GV20), Sishencong (EX-HN1), Fengchi (GB20), and Shenting (GV24), and RehaCom cognitive rehabilitation. The treatment duration was 8 weeks in both groups. Outcome measures were determined at baseline (week 0), 8 weeks after the first intervention (week 8; i.e., at the end of the intervention), and 12 weeks after completion of the intervention (week 20). The study design is summarized in Table 1.

### 2.2. Ethical Considerations

This study was conducted in accordance with the Declaration of Helsinki, and the protocol of this study (ver. 1.1) was approved by the Ministry of Food and Drug Safety (Medical Device Clinical Trial Plan approval number: 859; approval date: 24 July 2018) and the Institutional Review Board (IRB) of DongShin University Gwangju Korean Medicine Hospital (approval number: DSGOH-050; approval date: 17 September 2018) before the trial began. This trial was registered at the Clinical Research Information Service (cris.nih.go.kr; registration number: KCT0003415; registration date: 04 January 2019). The purpose and potential risks of this study were fully explained to the participants. All participants provided written informed consent before participating in this study.

### 2.3. Participant Recruitment

Participants were recruited at DongShin University Gwangju Korean Medicine Hospital. We submitted our study protocol to the Clinical Research Information Service on 12 November 2018. Considering the possibility of recruitment within the study period, we began the recruitment on 29 November 2018, which was before the trial registration. This study was advertised via local newspapers, the Internet, and posters in communities and hospitals. Participants received an explanation about the study from the clinical research coordinator (CRC) and were requested to voluntarily sign an informed consent form before participation. All recruited individuals were screened by the Korean version of the Mini-Mental State Examination (K-MMSE) and of the Montreal Cognitive Assessment (MoCA-K) to ensure that all inclusion criteria are met. The CRC monitored the medical conditions of the enrolled participants to maximize adherence to intervention protocols.

### 2.4. Participation

Participants who met all of the following criteria were included in the study: (1) Age 55–85 years; (2) fulfillment of the Peterson diagnostic criteria for MCI [1,2], with memory impairment for at least 3 months; (3) K-MMSE score of 20–23; (4) MoCA-K scale score of 0–22; (5) adequate Korean language fluency, for reliable completion of all study assessments; and (6) voluntary provision of informed consent.

The exclusion criteria were as follows: (1) Diagnosis of dementia according to the Diagnostic and Statistical Manual of Mental Disorders-IV; (2) history of structural brain lesions that could cause cognitive impairment, such as traumatic brain injury, stroke, intracranial space-occupying lesions, and congenital mental retardation; (3) presence of cancer and/or serious cardiovascular, cerebrovascular, liver, or kidney diseases; (4) history of treatment for alcohol or drug dependency or mental diseases, such as schizophrenia, serious anxiety, and depression in the past 6 months; (5) ongoing treatment for MCI, such as medication, acupuncture, and cognitive training); (6) difficulties in assessment due to visual and hearing impairments; (7) presence of contraindications for EA, such as blood clotting abnormalities (e.g., hemophilia), infection of the skin over the head, and presence of a pacemaker); and (8) concurrent participation in other clinical trials.

### 2.5. Randomization and Blinding

Following the acquisition of written informed consent, the practitioners who would perform the intervention conducted a screening interview. Thereafter, the assessor performed baseline measurements for the participants who met the inclusion criteria. The 36 enrolled participants were immediately assigned serial numbers generated using SPSS version 21 software (IBM Corp., Armonk, NY, USA) and were randomly allocated to one of the two study groups (n = 18 each group). The serial number codes were inserted into opaque envelopes that were sealed and kept in a double-locked cabinet; the envelopes were opened by the principal investigator or practitioner who would perform the intervention in the presence of the patient and a guardian.

We could only adopt a single outcome assessor-blinding approach because sham treatment was impossible because of the characteristics of EA application, which included insertion and electric stimulation. During the study, there was no contact between the assessor and any participant at any time point other than the time of assessment. Data analysts without conflicts of interest were involved in this study.

### 2.6. Implementation

The CRC generated the allocation sequence, enrolled the participants, and assigned participants to interventions.

### 2.7. Intervention

Participants in the CCR group received RehaCom cognitive rehabilitation (30 min) once a day, 3 days per week for 8 weeks. Participants in the EA-CCR group received EA (30 min) at Baihui (GV20), Sishencong (EX-HN1), Fengchi (GB20), and Shenting (GV24) in addition to RehaCom cognitive rehabilitation (30 min) once a day, 3 days per week for 8 weeks. EA was performed first, followed by CCR. The treatments were administered by Korean medicine doctors with 6 years of formal university training in Korean medicine and a license to administer treatment. To ensure strict adherence to the study protocol, the doctors received training together and used the same techniques.

#### 2.7.1. Electroacupuncture Treatment 

EA was performed at the following acupoints: Baihui (GV20), Sishencong (EX-HN1), Fengchi (GB20), and Shenting (GV24) [24]. Only sterile, stainless-steel, disposable acupuncture needles (size, 0.25 × 30 mm; Dong Bang Acupuncture, Inc., Boryeong, Republic of Korea; product no.: A84010.02) with guide tubes and an EA stimulator [CELLMAC PLUS (STN-330); Stratek, Co., Ltd., Anyang, Republic of Korea; product no.: A16010.04] were used. With the patients in the sitting position, the needles were inserted at an angle of 15–30° along the scalp. GB20 was punctured 17–30 mm in the direction of the tip of the nose. GV24, the anterior EX-HN1, and GV20 were punctured in the forward direction, while the left, right, and posterior EX-HN1 points were punctured in the direction of GV20. The depths of insertion were 9–24 mm, depending on the location of the needle [27]. After insertion, the needles were left in position for 30 min. Manual stimulation was not used. GV24 and GV20, the left and right EX-HN1, the anterior and posterior EX-HN1, and the left and right GB20 were subjected to EA under the following parameters: AC; continuous waves; frequency, 3 Hz; and intensity, between 2–4 mA such that the patient could feel it. Each participant received a total of 24 30-min sessions (three times per week for 8 weeks) [24] (Appendix A).

#### 2.7.2. RehaCom Cognitive Rehabilitation

All participants received RehaCom cognitive rehabilitation in the sitting position. Six different therapeutic programs to restore attention, memory, and executive functions were employed. Each program has one to four different tasks from which participants could choose during each therapy session. We mainly used topological memory, physiognomic memory, memory of words, and figural memory tasks of memory program and shopping, logical reasoning, and calculation tasks of executive function program. Each participant received a total of 24 30-min sessions (three times per week for 8 weeks).

During the clinical trial period, all participants were allowed to use routine management regimens, existing medications (e.g., those for hypertension, diabetes, or hyperlipidemia), and medications for maintaining and improving their health status. However, they were not permitted to engage in other treatments for ameliorating MCI symptoms. All medical devices, including the acupuncture needles, EA stimulator, and RehaCom software (HASOMED GmbH., Magdeburg, Germany), were inspected by the investigators, who recorded check-up results in the management register.

### 2.8. Outcome Measurements

Scores for the Korean version of Alzheimer’s Disease Assessment Scale—cognitive subscale (ADAS-K-cog), MoCA-K, Center for Epidemiological Studies—Depression Scale (CES-D), Korean Activities of Daily Living (K-ADL) scale, Korean Instrumental Activities of Daily Living (K-IADL) scale, and European Quality of Life Five Dimension Five Level Scale (EQ-5D-5L) were recorded before treatment, at the end of treatment, and at 12 weeks after treatment completion.

The primary outcome was improvement in cognitive function as assessed using the ADAS-K-cog, which is a tool that is considered the gold standard for assessing the efficacy of various anti-dementia treatments [28]. Particularly, it is known to be sensitive to the treatment responses of patients with MCI or early dementia [29]. 

The secondary outcomes included changes in the MoCA-K scale, CES-D, K-ADL scale, K-IADL scale, and EQ-5D-5L scores over time. The MoCA-K scale is a clinician-friendly, validated, brief instrument with high sensitivity and specificity for detecting MCI [30]. The CES-D is a short self-report scale designed to measure the current level of depressive symptomatology, with emphasis on the affective component (i.e., depressed mood) [31]. The K-ADL and K-IADL scales are used to assess physical function. The K-ADL scale is used to assess basic activities of elderly individuals; the K-IADL scale, to estimate complex activities representing instrumental self-maintenance and social behavior [32]. The EQ-5D-5L is a generic instrument for assessing health-related quality of life, which comprises five dimensions [33].

### 2.9. Sample Size Calculation

Because of the lack of adequate preliminary studies and limited research funds, study period, and recruitment opportunities, we have adopted a pilot study design with 18 participants in each group. As our study was a pilot study, the sample size was not sufficient to provide information on the efficacy of EA-CCR on MCI. Nevertheless, our study could provide an indication on the feasibility of a randomized trial of EA-CCR treatment for MCI and could determine whether EA-CCR is an acceptable treatment for patients with MCI.

### 2.10. Statistical Analyses

With the approval of the IRB, the statistical analysis in the study protocol was revised. We performed per-protocol (PP) analyses for the assessment of efficacy and a supplementary full analysis (FA) set. Missing values were imputed by the last observation carried forward method. We compared the results of the PP analyses and those of the supplementary FA set. If there was a significant difference between the PP and FA groups, the cause was reviewed and reflected during the efficacy assessment. Analysis was performed by blinded biostatisticians using SPSS version 20.0 software (SPSS Inc., Chicago, IL, USA); two-sided significance tests with a 5% significance level were employed. Continuous variables were presented as means and standard deviations and categorical variables as count frequencies and percentages.

Baseline data were obtained and compared using independent *t*-test, χ^2^ test, and Fisher’s exact test. Differences in all outcome value changes in the two groups were compared using Wilcoxon signed-rank test and repeated-measures analysis of variance (ANOVA; Friedman tests). ADAS-K-cog, MoCA-K, CES-D, K-ADL scale, K-IADL scale, and EQ-5D-5L values were compared by repeated-measures ANOVA across two to three testing time points (i.e., week 0, week 8, and week 20). Differences in outcome value changes between the two groups (week 0 vs. week 8, week 0 vs. week 20, and week 8 vs. week 20) were compared using the Mann–Whitney U-test (nonparametric test). Moreover, the participants were divided into two groups according to age: <70 and >70 groups, and a subanalysis was conducted to investigate the differences in ADAS-K-cog, MoCA-K, CES-D, K-ADL scale, K-IADL scale, and EQ-5D-5L changes (week 0 vs. week 8, week 0 vs. week 20, and week 8 vs. week 20) between the two groups.

## 3. Results

### 3.1. Particpants

We recruited participants between 29 November 2018 and 23 October 2019. Of the 476 patients assessed for eligibility, 440 were excluded. Thirty-six patients were included in this study and were randomly assigned to either the EA-CCR group (*n* = 18) or CCR group (*n* = 18). Two participants in both groups did not complete the treatment. Results of the PP analysis for the assessment of efficacy were not different from those of the supplementary FA set. Thus, data of 32 patients with MCI (EA-CCR group, *n* = 16; CCR group, *n* = 16) were used in the final analysis (Figure 1).

### 3.2. Baseline Charactersitics

The baseline demographic characteristics and study variables of the 32 patients in the two groups are presented in Table 2. No significant differences in the baseline demographic characteristics and study variables were detected between the two groups (*p >* 0.05; Table 2).

### 3.3. Primary and Secondary Outcomes

After eight weeks of intervention, we observed significant improvements in both groups (changes in ADAS-K-cog and MoCA-K) and in the EA-CCR group (changes in CES-D) (Table 3).

Repeated-measures ANOVA showed no significant interaction between time and group with respect to all study variables (Table 4).

No significant differences in the changes in ADAS-K-cog, MoCA-K, CES-D, K-ADL, K-IADL, and EQ-5D-5L (week 0 vs. week 8, week 0 vs. week 20, and week 8 vs. week20) were found between the two groups (Table 5).

A subanalysis based on patient age (i.e., <70 years and >70 years) showed no significant differences in all variables between the two groups (Table 6 and Table 7).

### 3.4. Safety Evaluation

Adverse events in this study were recorded on a case report form, and their relationship with the intervention was evaluated. No adverse events related to the intervention occurred in this study.

## 4. Discussion

To the best of our knowledge, this is the first randomized controlled study to investigate the effects of EA-CCR on cognitive function, depression, activities of daily living, and quality of life in patients with MCI by comparing the effects of EA-CCR with those of CCR alone. Our study design, which includes eight weeks of treatment [16,24], specific acupoints for acupuncture [16,17,24], and EA treatment method [24], was based on a previous study. 

We observed significant improvements in both groups (i.e., changes in ADAS-cog and MoCA-K) and in EA-CCR group (i.e., changes in CES-D). However, EA in EA-CCR showed no positive add-on effects on cognitive function, depression, activities of daily living, and quality of life in patients with MCI. A subanalysis according to age also demonstrated no positive add-on effects of EA.

Systematic reviews reported that both EA and CCR could improve cognitive function and thus are effective treatments for patients with MCI [12,17]. However, in the EA-CCR group, no positive add-on cognitive improvement effects were observed. We postulate several reasons for our results. First, the small sample size, inclusion criteria, and treatment frequency possibly influenced the results. MCI is a neurodegenerative disease that is slowly progressive; thus, eight weeks of EA may not be enough to improve cognitive function. The EA-CCR group (n = 18) in our study received a total of 24 30-min sessions (once daily, three times per week for eight weeks). In a previous study that showed positive synergistic effects of acupuncture and CCR on cognitive function improvement, the combination treatment group (poststroke patients, n = 60) received a total of 60 30-min sessions (once daily, 5 days per week for 12 weeks) [34]. Second, the intervention that is combined with EA may affect the results. In previous studies that showed positive add-on effects of acupuncture on MCI [35], EA was combined with pharmacological treatment. In our study, we used CCR with EA to investigate the effects of combinational treatment using non-pharmacologic interventions on MCI. The interaction between EA and CCR may affect the results. Third, acupoint specificity may have affected the results. The selection and compatibility of acupoints have a direct effect on therapeutic effects. According to the concept of “holism” in traditional Chinese medicine, acupoints in limbs, especially those located below the elbow and knee joints, are extremely important for managing organ and meridian diseases. These points could be therapeutic for local and systemic problems [36]. In another systematic review that reported the cognitive improvement effects of EA in patients with MCI, four of the five studies used both scalp and body acupuncture and one study used scalp acupuncture only [17]. In our study, we used scalp acupuncture only based on a previous study [24].

Our study has some limitations. First, we adopted a single outcome assessor-blinded approach because sham treatment was impossible given the characteristics of EA application. This limitation may have led to a bias in the results of the study. Second, because of limited research funds, study period, and recruitment opportunities, our study did not have enough sample size and long follow-up period to investigate the cognitive improvement effects and long-term effects of EA on MCI. This limitation also may have affected the results of this study. Thus, it is necessary to conduct further studies with enough sample size and long follow-up period to investigate the cognitive improvement effects and long-term effects of EA on MCI. Third, we did not investigate the add-on effect of EA through various acupuncture methods. Apart from needle insertion, issues such as needling sensation, psychological factors, acupoint specificity, acupuncture manipulation, and needle duration also have relevant influences on the therapeutic effects of acupuncture [37]. While several different acupuncture methods for treating MCI exist, we only performed EA at Baihui (GV20), Sishencong (EX-HN1), Fengchi (GB20), and Shenting (GV24) for 30 min. Thus, further studies on effective acupuncture methods are warranted. Fourth, this study was a pilot study to investigate the cognitive improvement effects of EA-CCR, so we did not evaluate the cost-effectiveness of EA-CCR. Thus, further studies on cost-effectiveness of EA-CCR are needed.

## 5. Conclusions

Results in our study indicate that EA-CCR and CCR have beneficial effects on cognitive function improvement in patients with MCI. However, EA-CCR did not show positive add-on effects of EA on the improvement of cognitive function, depression, activities of daily living, and quality of life in patients with MCI. Moreover, no significant differences in outcomes between the two treatments were noted.

Nevertheless, we believe that the results of our study could have varied greatly depending on sample size, frequency and total number of sessions, intervention that is combined with EA, and acupoint specificity. We hope that well-designed RCTs with enough sample size aimed at investigating possible effects or add-on effects of EA on MCI will be conducted in the future.

## Figures and Tables

**Figure 1 brainsci-10-00984-f001:**
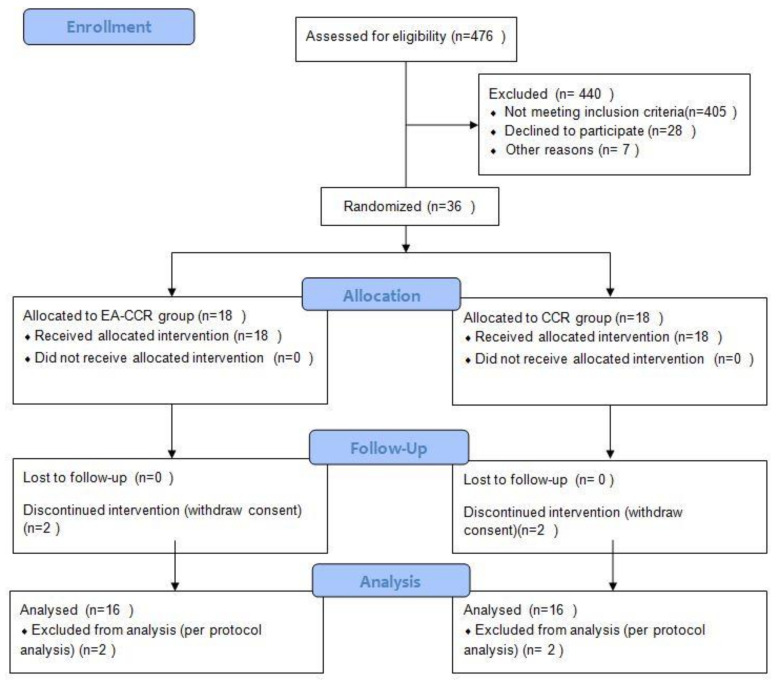
CONSORT 2010 flow diagram.

**Table 1 brainsci-10-00984-t001:** Standard protocol items: Recommendations for Interventional Trials (SPIRIT) statement.

	STUDY PERIOD										
	Enrolment	Allocation	Post-Allocation								Close-Out
**TIMEPOINT**	**Screening**		Visit1-3	Visit4-6	Visit7-9	Visit10-12	Visit13-15	Visit16-18	Visit19-21	Visit22-24	Visit25
	Week		1	2	3	4	5	6	7	8	20
ENROLMENT											
Informed consent	X										
Sociodemographic profile	X										
Medical history	X										
Vital signs	X	X	X	X	X	X	X	X	X	X	X
Inclusion/exclusion criteria	X										
Allocation		X									
K-MMSE, MoCA-K	X										
INTERVENTIONS											
CCR			X	X	X	X	X	X	X	X	
EA-CCR			X	X	X	X	X	X	X	X	
ASSESSMENTS											
Change of medical history			X	X	X	X	X	X	X	X	X
Safety assessment			X	X	X	X	X	X	X	X	X
ADAS-K-cog			X							X	X
MoCA-K			X							X	X
CES-D			X							X	X
K-ADL, K-IADL			X							X	X
EQ-5D-5L			X							X	X

The table shows the enrollment, interventions, and data collection protocols. K-MMSE, Korean version of Mini-Mental State Examination; MoCA-K, Korean version of the Montreal Cognitive Assessment; CCR, computer-based cognitive rehabilitation; EA-CCR, electroacupuncture combined with computer-based cognitive rehabilitation; ADAS-K-cog, Korean version of Alzheimer’s Disease Assessment Scale—cognitive subscale; CES-D, Center for Epidemiological Studies—Depression Scale; K-ADL, Korean Activities of Daily Living; K-IADL Korean Instrumental Activities of Daily Living; EQ-5D-5L, European Quality of Life Five Dimension-Five Level Scale.

**Table 2 brainsci-10-00984-t002:** Homogeneity tests for baseline demographic characteristics and study variables for 32 patients with Mild Cognitive Impairment.

Dependent Variables	EA-CCR (*n* = 16)	CCR (*n* = 16)	*p* or *x*^2^ *(P)*
	Mean (SD) or n (%)	Mean (SD) or n (%)	
Age (y)	69.94 (5.94)	74.25 (5.39)	21.33 (0.166) ^ǂ^
Gender (Female)	14 (87.5)	14 (87.5)	0.00 (1.000) ^ǂ^
Education	9.13 (4.83)	8.69 (5.08)	4.84 (0.679) ^ǂ^
ADAS-K-cog	11.13 ± 4.10	11.19 ± 6.22	−0.03 (0.973) *
MoCA-K	18.75 ± 2.54	19.31 ± 2.92	−0.58 (0.565) *
CES-D	14.25 ± 5.91	11.50 ± 6.94	1.21 (0.237) *
K-ADL	7.19 ± 0.40	7.13 ± 0.34	0.47 (0.640) *
K-IADL	11.00 ± 3.03	11.50 ± 3.10	−0.46 (0.648) *
EQ-5D-5L	6.38 ± 1.36	7.31 ± 2.52	−1.31 (0.201) *

* *t*-test; ^ǂ^
*x*^2^-test.

**Table 3 brainsci-10-00984-t003:** Changes in outcome measures (week 0 vs. week 8, week 0 vs. week 20) after treatment completion in patients who received electroacupuncture combined with computer-based cognitive rehabilitation (EA-CCR) and computer-based cognitive rehabilitation (CCR) (n = 16 each) for Mild Cognitive Impairment.

Groups	Dependent Variables	Week 0 (M ± SD)	Week 8 (M ± SD)	Week 20 (M ± SD)	Difference (w8-w0)	Z (*p*) *	Difference (w20-w0)	Z (*p*) *	x^2^ (*p*) ǂ
EA-CCR group (*n* = 16)	ADAS-K -cog	11.13 ± 4.10	7.19 ± 4.75	6.19 ± 3.95	−3.94 ± 2.57	−3.42 (0.001)	−4.94 ± 3.45	−3.42 (0.001)	21.08 (<0.001)
	MoCA-K	18.75 ± 2.54	24.25 ± 3.26	24.56 ± 4.21	5.50 ± 2.48	−3.53 (<0.001)	5.81 ± 3.69	−3.41 (0.001)	24.10 (<0.001)
	CES-D	14.25 ± 5.91	10.94 ± 6.18	10.75 ± 6.02	−3.31 ± 4.77	−2.39 (0.017)	−3.50 ± 6.20	−2.01 (0.038)	7.48 (0.024)
	K-ADL	7.19 ± 0.40	7.06 ± 0.25	7.13 ± 0.34	−0.13 ± 0.34	−1.41 (0.157)	−0.06 ± 0.44	−0.58 (0.564)	2.00 (0.368)
	K-IADL	11.00 ± 3.03	10.69 ± 1.85	10.44 ± 0.96	−0.31 ± 3.03	−0.54 (0.593)	−0.56 ± 2.45	−0.73 (0.465)	0.15 (0.926)
	EQ-5D-5L	6.38 ± 1.36	6.06 ± 1.24	5.75 ± 1.00	0.31 ± 1.49	−0.93 (0.351)	0.63 ± 1.54	−1.40 (0.161)	3.56 (0.169)
CCR group (*n* = 16)	ADAS-K -cog	11.19 ± 6.22	7.38 ± 3.86	5.81 ± 2.76	-3.81 ± 3.47	−3.33 (0.001)	−5.38 ± 5.19	−3.42 (0.001)	19.22 (<0.001)
	MoCA-K	19.31 ± 2.92	24.19 ± 2.48	25.13 ± 1.89	4.88 ± 2.45	−3.53 (<0.001)	5.81 ± 2.37	−3.53 (<0.001)	26.00 (<0.001)
	CES-D	11.50 ± 6.94	11.00 ± 6.69	9.75 ± 6.77	−0.50 ± 7.43	−0.00 (1.00)	−1.75 ± 8.70	−1.02 (0.306)	1.86 (0.395)
	K-ADL	7.13 ± 0.34	7.06 ± 0.25	7.06 ± 0.25	−0.06 ± 0.25	−1.00 (0.317)	−0.06 ± 0.25	−1.00 (0.317)	2.00 (0.368)
	K-IADL	11.50 ± 3.10	11.44 ± 3.08	11.62 ± 3.12	−0.06 ± 0.93	−0.45 (0.655)	−0.13 ± 1.31	−0.54 (0.593)	0.20 (0.905)
	EQ-5D-5L	7.31 ± 2.52	6.12 ± 1.24	6.31 ± 1.25	1.00 ± 2.92	−2.21 (0.027)	−1.00 ± 2.92	−1.29 (0.196)	5.28 (0.071)

* Wilcoxon signed-rank test; ^ǂ^ Repeated measures ANOVA (Friedman test).

**Table 4 brainsci-10-00984-t004:** Results of repeated-measures ANOVA for the outcomes of treatment between patients who received EA-CCR and CCR for Mild Cognitive Impairment (*n* = 16 each).

Dependent Variables	Group (n)	Week 0 (M ± SD)	Week 8 (M ± SD)	Week 20 (M ± SD)	Source	SS	df	Mean Square	F	*p*
ADAS-K-cog	EA-CCR (*n* = 16)	11.13 ± 4.10	7.19 ± 4.75	6.19 ± 3.95	Time	461.27	2	230.64	38.16	<0.001
	CCR (*n* = 16)	11.19 ± 6.22	7.38 ± 3.86	5.81 ± 2.76	Group Time	1.396	2	0.70	0.12	0.891
MoCA-K	EA-CCR (*n* = 16)	18.75 ± 2.54	24.25 ± 3.26	24.56 ± 4.21	Time	651.58	2	325.79	86.11	<0.001
	CCR (*n* = 16)	19.31 ± 2.92	24.19 ± 2.48	25.13 ± 1.89	Group Time	2.08	2	1.04	0.28	0.760
CES-D	EA-CCR (*n* = 16)	14.25 ± 5.91	10.94 ± 6.18	10.75 ± 6.02	Time	117.77	2	58.89	2.52	0.089
	CCR (*n* = 16)	11.50 ± 6.94	11.00 ± 6.69	9.75 ± 6.77	Group Time	32.27	2	16.14	0.69	0.506
K-ADL	EA-CCR (*n* = 16)	7.19 ± 0.40	7.06 ± 0.25	7.13 ± 0.34	Time	0.15	2	0.07	1.75	0.183
	CCR (*n* = 16)	7.13 ± 0.34	7.06 ± 0.25	7.06 ± 0.25	Group Time	0.02	2	0.01	0.25	0.780
K-IADL	EA-CCR (*n* = 16)	11.00 ± 3.03	10.69 ± 1.85	10.44 ± 0.96	Time	0.90	2	0.45	0.26	0.770
	CCR (*n* = 16)	11.50 ± 3.10	11.44 ± 3.08	11.62 ± 3.12	Group Time	1.94	2	0.97	0.57	0.570
EQ-5D-5L	EA-CCR (*n* = 16)	6.38 ± 1.36	6.06 ± 1.24	5.75 ± 1.00	Time	13.08	2	6.54	3.83	0.027
	CCR (*n* = 16)	7.31 ± 2.52	6.12 ± 1.24	6.31 ± 1.25	Group Time	3.08	2	1.54	0.90	0.411

**Table 5 brainsci-10-00984-t005:** Comparison of changes in outcome measurements between patients who received EA-CCR and CCR for Mild Cognitive Impairment (*n* = 16 each).

Dependent Variables	Group(n)	Week 0 (M ± SD)	Difference (w8-w0)	*Z* (*p*) *	Difference (w20-w0)	*Z* (*p*) *	Difference (w20-w8)	*Z* (*p*) *
ADAS-K -cog	EA-CCR (*n* = 16)	11.13 ± 4.10	−3.94 ± 2.57	−0.38 (0.703)	−4.94 ± 3.45	−0.21 (0.835)	−1.00 ± 2.66	−0.25 (0.804)
	CCR(n = 16)	11.19 ± 6.22	−3.81 ± 3.47		−5.38 ± 5.19		1.56 ± 2.83	
MoCA-K	EA-CCR (*n* = 16)	18.75 ± 2.54	5.50 ± 2.48	−0.72 (0.470)	5.81 ± 3.69	−0.23 (0.819)	0.31 ± 2.98	−0.74 (0.459)
	CCR(n = 16)	19.31 ± 2.92	4.88 ± 2.45		5.81 ± 2.37		0.94 ± 2.26	
CES-D	EA-CCR (*n* = 16)	14.25 ± 5.91	−3.31 ± 4.77	−1.32 (0.186)	−3.50 ± 6.20	−0.63 (0.533)	−0.19 ± 4.05	−1.19 (0.234)
	CCR(n = 16)	11.50 ± 6.94	−0.50 ± 7.43		−1.75 ± 8.70		−1.25 ± 8.50	
K-ADL	EA-CCR (*n* = 16)	7.19 ± 0.40	−0.13 ± 0.34	−0.60 (0.551)	−0.06 ± 0.44	−0.03 (0.974)	0.06 ± 0.25	−1.00 (0.317)
	CCR(n = 16)	7.13 ± 0.34	−0.06 ± 0.25		−0.06 ± 0.25		0.00 ± 0.00	
K-IADL	EA-CCR (*n* = 16)	11.00 ± 3.03	−0.31 ± 3.03	−0.45 (0.655)	−0.56 ± 2.45	−0.42 (0.677)	−0.25 ± 1.29	−0.07 (0.948)
	CCR(n = 16)	11.50 ± 3.10	−0.06 ± 0.93		−0.13 ± 1.31		0.19 ± 1.05	
EQ−5D−5L	EA-CCR (*n* = 16)	6.38 ± 1.36	0.31 ± 1.49	−1.31 (0.189)	0.63 ± 1.54	−0.21(0.832)	−0.31 ± 0.95	−1.17(0.243)
	CCR(*n* = 16)	7.31 ± 2.52	−1.19 ± 1.91		−1.00 ± 2.92		0.19 ± 1.68	

* Mann-Whitney U-test.

**Table 6 brainsci-10-00984-t006:** Comparison of changes in outcome measurement between patients who received EA-CCR (*n* = 9) and CCR (*n* = 4) for under 70 years old group.

Dependent Variables	Group (*n*)	Week 0 (M ± SD)	Week 8 (M ± SD)	Week 20 (M ± SD)	Difference (w8-w0)	*Z* (*p*) *	Difference (w20-w0)	*Z* (*p*) *	Difference (w20-w8)	*Z* (*p*) *
ADAS-cog	EA-CCR (*n* = 9)	11.56 ± 3.94	7.56 ± 5.66	7.00 ± 4.64	−4.00 ± 3.24	−0.70 (0.483)	−4.56 ± 4.03	−0.94 (0.349)	−0.56 ± 2.74	−0.80 (0.424)
	CCR (*n* = 4)	7.00 ± 2.94	4.75 ± 3.40	4.75 ± 3.40	−2.25 ± 1.71		−2.25 ± 0.96		0.00 ± 2.00	
MoCA-K	EA-CCR (*n* = 9)	18.33 ± 2.50	23.44 ± 3.47	23.00 ± 4.33	5.11 ± 2.57	−0.47 (0.639)	4.67 ± 3.24	−1.02 (0.308)	−0.44 ± 2.88	−0.78 (0.438)
	CCR (*n* = 4)	19.75 ± 2.22	25.50 ± 2.52	26.25 ± 1.71	5.75 ± 2.06		6.50 ± 1.00		0.75 ± 1.26	
CES-D	EA-CCR (*n* = 9)	13.78 ± 4.47	11.22 ± 6.24	12.22 ± 4.32	−2.56 ± 3.64	−0.16 (0.876)	−1.56 ± 5.27	−0.70 (0.486)	1.00 ± 4.72	−0.46 (0.643)
	CCR (*n* = 4)	10.50 ± 7.59	11.50 ± 9.29	13.00 ± 7.75	1.00 ± 8.68		2.50 ± 14.57		1.50 ± 16.11	
K-ADL	EA-CCR (*n* = 9)	7.00 ± 0.00	7.00 ± 0.00	7.11 ± 0.33	0.00 ± 0.00	0.00 (1.000)	0.11 ± 0.33	−0.67 (0.505)	0.11 ± 0.33	−0.68 (0.505)
	CCR (*n* = 4)	7.00 ± 0.00	7.00 ± 0.00	7.00 ± 0.00	0.00 ± 0.00		0.00 ± 0.00		0.00 ± 0.00	
K-IADL	EA-CCR (*n* = 9)	11.78 ± 3.96	11.22 ± 2.39	10.78 ± 1.20	−0.56 ± 4.13	−0.42 (0.676)	−1.00 ± 3.28	−0.79 (0.429)	−0.44 ± 1.74	−0.94 (0.348)
	CCR (*n* = 4)	10.75 ± 1.50	10.75 ± 1.50	11.75 ± 2.06	0.00 ± 0.00		1.00 ± 2.00		1.00 ± 2.00	
EQ−5D−5L	EA-CCR (*n* = 9)	6.44 ± 1.59	5.78 ± 1.09	5.33 ± 0.71	−0.67 ± 1.41	−0.24 (0.810)	−1.11 ± 1.69	−0.24 (0.813)	−0.44 ± 1.13	−0.50 (0.614)
	CCR (*n* = 4)	6.75 ± 2.06	6.00 ± 1.41	6.50 ± 1.73	−0.75 ± 0.96		−0.25 ± 3.10		0.50 ± 2.52	

* Mann-Whitney U-test.

**Table 7 brainsci-10-00984-t007:** Comparison of changes in outcome measurements between patients who received EA-CCR (*n* = 7) and CCR (*n* = 12) for over 70 years old group.

Dependent Variables	Group (*n*)	Week 0 (M ± SD)	Week 8 (M ± SD)	Week 20 (M ± SD)	Difference (w8-w0)	*Z* (*p*) *	Difference (w20-w0)	*Z* (*p*) *	Difference (w20-w8)	*Z* (*p*) *
ADAS-cog	EA-CCR (*n* = 7)	10.57 ± 4.54	6.71 ± 3.64	5.14 ± 2.85	−3.86 ± 1.57	−0.86 (0.931)	−5.43 ± 2.76	−0.85 (0.932)	−1.57 ± 2.64	−0.43 (0.668)
	CCR (*n* = 12)	12.58 ± 6.47	8.25 ± 3.72	6.17 ± 2.55	−4.33 ± 3.80		−6.42 ± 5.63		−2.08 ± 2.94	
MoCA-K	EA-CCR (*n* = 7)	19.29 ± 2.69	25.29 ± 2.87	26.57 ± 3.31	6.00 ± 2.45	−1.41 (0.158)	7.29 ± 3.95	−1.11 (0.265)	1.29 ± 3.04	−0.86 (0.932)
	CCR (*n* = 12)	19.17 ± 3.19	23.75 ± 2.42	24.75 ± 1.86	4.58 ± 2.57		5.58 ± 2.68		1.00 ± 2.56	
CES-D	EA-CCR (*n* = 7)	14.86 ± 7.73	10.57 ± 6.58	8.86 ± 7.63	−4.29 ± 6.10	−1.32 (0.188)	−6.00 ± 6.78	−0.93 (0.352)	−1.71 ± 2.56	−0.55 (0.579)
	CCR (*n* = 12)	11.83 ± 7.03	10.83 ± 6.12	8.67 ± 6.40	−1.00 ± 7.32		−3.17 ± 6.04		−2.17 ± 4.91	
K-ADL	EA-CCR (*n* = 7)	7.43 ± 0.53	7.14 ± 0.38	7.14 ± 0.38	−0.29 ± 0.49	1.14 (0.256)	−0.29 ± 0.49	−1.14 (0.258)	0.00 ± 0.00	−0.00 (1.000)
	CCR (*n* = 12)	7.17 ± 0.39	7.08 ± 0.29	7.08 ± 0.29	−0.08 ± 0.29		−0.08 ± 0.29		0.00 ± 0.00	
K-IADL	EA-CCR (*n* = 7)	10.00 ± 0.00	10.00 ± 0.00	10.00 ± 0.00	0.00 ± 0.00	−0.00 (1.000)	0.00 ± 0.00	−0.00 (1.000)	−0.00 ± 0.00	−0.76 (0.445)
	CCR (*n* = 12)	11.75 ± 3.49	11.67 ± 3.47	11.58 ± 3.48	−0.08 ± 1.08		−0.17 ± 0.94		−0.08 ± 0.29	
EQ−5D−5L	EA-CCR (*n* = 7)	6.29 ± 1.11	6.43 ± 1.40	6.29 ± 1.11	0.14 ± 1.57	−1.45 (0.146)	0.00 ± 1.15	−0.78 (0.433)	−0.14 ± 0.69	−0.83 (0.409)
	CCR (*n* = 12)	7.50 ± 2.71	6.17 ± 1.19	6.25 ± 1.14	−1.33 ± 2.15		−1.25 ± 2.96		0.08 ± 1.44	

* Mann-Whitney U-test.

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
