# Peer review of "Cognitive Improvement Effects of Electroacupuncture Combined with Computer-Based Cognitive Rehabilitation in Patients with Mild Cognitive Impairment: A Randomized Controlled Trial"

_brainsci, 2020, doi:10.3390/brainsci10120984_

Round 1

Reviewer 1 Report

Review of Cognitive improvement effects of electroacupuncture combined with computer-based cognitive rehabilitation in patients with mild cognitive impairment: a randomized controlled trial

The manuscript is well written and clearly explains most points. Given that this is presented as a pilot study, with a very low N, it seems reasonable to treat it with some latitude, since the authors clearly identified the limitations of the study. The comments I will offer below are suggestions for improving the clarity of the paper. I do not see these as procedural or analytical problems.

1.0 The research was apparently inspired by other studies (referenced) that suggested possible success with EA. It would help to offer a short explanation of what is believed to be the reason that EQ might offer a benefit not seen with acupuncture alone. For example, is there reason to believe that the electrical stimulation interacts with the cortex or other parts of the brain in such a way as to cause physical changes in the brain? Even if this is discussed in the references, a short summary would help some readers. [lines 55,56]

2.0 Duration - This study was set for an 8 week duration. [line 77] The paper should address these:
Was that time selected on the basis of other treatment studies, limitation of funding, or other reasons?
Is the null outcome possibly the result of a treatment period that is not long enough?

3.0 Cost efficiency - If it is possible to determine the cost effectiveness of treatment of the CCR group, is it possible to extrapolate an expected cost for EA-CCR? For example, it appears that the EA-CCR took about twice as much time as the CCR alone. I realize that the null finding may make it unproductive to estimate EA-CCR value. Perhaps a statement could be offered to explain how much extra gain would be required to go from CCR to EA-CCR.

4.0 There was no mention of fadeout. If the literature addresses fadeout, that should be mentioned; if it does not, a discussion of how to deal with it should be included. Most of the work done on training related to intelligence has shown relatively rapid fadeout with little or no long term effects. The question is whether one can reasonably expect this sort of treatment (if the results were reasonably strong) to produce lasting results, or would it require continuous treatment? One way of addressing this issue is to simply mention how future research should be conducted to evaluate fadeout.

5.0 Electrical stimulation - [line 164] A fluctuating electrical signal can be applied with only direct current that is changing in voltage, but always with the same polarity. Or, an AC signal can be applied with alternating polarity. It is not clear as to which was used. The reader is likely to guess AC. Why was the frequency varying? Is this the range of stability of the equipment, plus individual differences, or was the frequency deliberately altered over the range stated? Similarly, was the current range a matter of differences between individuals, or a natural instability in the process? Are these electrical parameters standard for similar EA work? These and related comments do not have to be resolved with this reviewer; they are suggestions. One way to address them is to include short discussions in an appendix, footnote, or endnote.

6.0 RehaCom - [lines 166 - 170] A brief (one or two sentences) description of the tasks would be helpful to anyone who is not familiar with ReahCom. I assume most readers will already know this software.

7.0 Sex distribution - The study was strongly skewed towards female participants. I assume this was a matter of chance, but it might be worth commenting on whether the EA-CCR is likely (if a strong result were found) to show a sex difference in effectiveness. Perhaps suggestions for future research could address this. If the question is not yet answerable, future researchers could be advised that a balance of male and female participants would be useful in detecting sex differences, if they exist.

8.0 Null finding - [line 278] The comments relating to the lack of an EA related effect are appropriate and realistic, given the small N. The prior study mentioned [line 287] seems more encouraging for future research than this pilot study. There were procedural differences mentioned: a more intense treatment over a longer time span; the use of pharmacological agents; and possible differences in acupoint specificity. It makes sense to test without the pharmacological agents (as this study did). The pilot study suggests that the pharmacological agents may have been responsible for prior results, but there is the problem of the very limited scope of this pilot study. I suggest a slightly expanded discussion of this section.

9.0 Future studies - [line 320+] I think this should be the most important part of the paper. If the authors believe that more work should be done with EA, it would be helpful to make a strong case for such study. The null finding does not seem encouraging, so a more detailed discussion of why one should spend research funds on continuing the work requires a more detailed recommendation. Among the things that might be helpful to someone considering future work:

- Should another pilot study be done, or should a large program be done. If a pilot program is recommended, what is the minimum number of cohorts that should be considered?
- What frequency and duration of treatments should be adopted?
- How long should the study last?
- Should there be a balanced male and female participation?
- Should the study include an EA-only group?

Author Response

We are very grateful for the constructive comments and the opportunity to revise our manuscript.

Point-by-point responses to the reviewers’ comments are provided below.

The manuscript is well written and clearly explains most points. Given that this is presented as a pilot study, with a very low N, it seems reasonable to treat it with some latitude, since the authors clearly identified the limitations of the study. The comments I will offer below are suggestions for improving the clarity of the paper. I do not see these as procedural or analytical problems.

Point 1: 1.0 The research was apparently inspired by other studies (referenced) that suggested possible success with EA. It would help to offer a short explanation of what is believed to be the reason that EQ might offer a benefit not seen with acupuncture alone. For example, is there reason to believe that the electrical stimulation interacts with the cortex or other parts of the brain in such a way as to cause physical changes in the brain? Even if this is discussed in the references, a short summary would help some readers. [lines 55,56]

Response 1: Thank you for your valuable comments. In accordance with your suggestion, we have inserted the pertinent sentence in the Introduction section as follows: “EA has been reported to produce greater effect on neuroblast plasticity in the dentate gyrus [18], more widespread signal increases in the human brain as measured by functional magnetic resonance imaging [19] than acupuncture alone.” (page 2, line 55-57)

Point 2: Duration - This study was set for an 8 week duration. [line 77] The paper should address these: Was that time selected on the basis of other treatment studies, limitation of funding, or other reasons? Is the null outcome possibly the result of a treatment period that is not long enough?

Response 2: Thank you for your query. We selected 8 weeks of treatment based on the previous study. Four RCTs in the meta-analysis paper on acupuncture treatment for MCI selected an eight-week treatment period [1]. We have inserted the pertinent sentence in the Discussion section as follows: “Our study design, which includes 8 weeks of treatment [16,24], specific acupoints for acupuncture [16,17,24] and EA treatment method [24], was based on a previous study.” (page 11, line 283-285)

Point 3: Cost efficiency - If it is possible to determine the cost effectiveness of treatment of the CCR group, is it possible to extrapolate an expected cost for EA-CCR? For example, it appears that the EA-CCR took about twice as much time as the CCR alone. I realize that the null finding may make it unproductive to estimate EA-CCR value. Perhaps a statement could be offered to explain how much extra gain would be required to go from CCR to EA-CCR.

Response 3: Thank you for your valuable comments. In accordance with your suggestion, we have inserted the pertinent sentence in the Discussion section as follows: “Fourth, this study was a pilot study to investigate the cognitive improvement effects of EA-CCR, so we did not evaluate the cost-effectiveness of EA-CCR. Thus further studies on cost-effectiveness of EA-CCR are needed.” (page 12, line 324-326)

Point 4 : There was no mention of fadeout. If the literature addresses fadeout, that should be mentioned; if it does not, a discussion of how to deal with it should be included. Most of the work done on training related to intelligence has shown relatively rapid fadeout with little or no long term effects. The question is whether one can reasonably expect this sort of treatment (if the results were reasonably strong) to produce lasting results, or would it require continuous treatment? One way of addressing this issue is to simply mention how future research should be conducted to evaluate fadeout.

Response 4: Thank you for your valuable comments. In accordance with your suggestion, we have inserted the pertinent sentence in the Discussion section as follows: “Second, because of limited research funds, study period, and recruitment opportunities, our study did not have enough sample size and long follow-up period to investigate the cognitive improvement effects and long term effects of EA on MCI. This limitation also may have affected the results of this study. Thus, it is necessary to conduct further studies with enough sample size and long follow-up period to investigate the cognitive improvement effects and long term effects of EA on MCI.” (page 11, line 313-318)

Point 5: Electrical stimulation - [line 164] A fluctuating electrical signal can be applied with only direct current that is changing in voltage, but always with the same polarity. Or, an AC signal can be applied with alternating polarity. It is not clear as to which was used. The reader is likely to guess AC. Why was the frequency varying? Is this the range of stability of the equipment, plus individual differences, or was the frequency deliberately altered over the range stated? Similarly, was the current range a matter of differences between individuals, or a natural instability in the process? Are these electrical parameters standard for similar EA work? These and related comments do not have to be resolved with this reviewer; they are suggestions. One way to address them is to include short discussions in an appendix, footnote, or endnote.

Response 5: Thank you for your valuable comments. We are sorry to confuse you. It was our mistake. We used AC. The frequency was not varying. The frequency was 3 Hz and we adjusted the intensity of the current between 2 and 4 mA to the extent that the patient could feel it. We revised the ‘Electroacupuncture treatment’ subsection for more clarity as follow : “GV24 and GV20, the left and right EX-HN1, the anterior and posterior EX-HN1, and the left and right GB20 were subjected to EA under the following parameters: AC; continuous waves; frequency, 3 Hz; and intensity, 2–4 mA to the extent that the patient could feel it.” (page 5, line 168-170)

Point 6: RehaCom - [lines 166 - 170] A brief (one or two sentences) description of the tasks would be helpful to anyone who is not familiar with ReahCom. I assume most readers will already know this software.

Response 6: Thank you for your vaulable comments. In accordance with your suggestions, we have added a statement in the Rehacom cognitive rehabilitation subsection as follows: “We mainly used topological memory, physiognomic memory, memory of words, and figural memory tasks of memory program and shopping, logical reasoning, and calculation tasks of executive function program.” (page 5, line 176-178)

Point 7 : Sex distribution - The study was strongly skewed towards female participants. I assume this was a matter of chance, but it might be worth commenting on whether the EA-CCR is likely (if a strong result were found) to show a sex difference in effectiveness. Perhaps suggestions for future research could address this. If the question is not yet answerable, future researchers could be advised that a balance of male and female participants would be useful in detecting sex differences, if they exist.

Response 7: Thank you for your comments. Our study was strongly skewed towards female participants. However, there was no significant difference in gender ratio between the two groups. To the best of our knowledge, there have been no paper to report on gender differences in effects of cognitive rehabilitation treatment. We think it is unlikely that gender affected the results of our study. However, further study will require a sub-analysis based on participant gender or balance of male and female participants.

Point 8 : Null finding - [line 278] The comments relating to the lack of an EA related effect are appropriate and realistic, given the small N. The prior study mentioned [line 287] seems more encouraging for future research than this pilot study. There were procedural differences mentioned: a more intense treatment over a longer time span; the use of pharmacological agents; and possible differences in acupoint specificity. It makes sense to test without the pharmacological agents (as this study did). The pilot study suggests that the pharmacological agents may have been responsible for prior results, but there is the problem of the very limited scope of this pilot study. I suggest a slightly expanded discussion of this section.

Response 8: Thank you for your valuable comments. In accordance with your suggestions, we have added a statement in the Discussion section as follows: “Second, the intervention that is combined with EA may affect the results. In previous studies that showed positive add-on effects of acupuncture on MCI [37], EA was combined with pharmacological treatment. In our study, we used CCR with EA to investigate the effects of combinational treatment using non-pharmacologic interventions on MCI. The interaction between EA and CCR may affect the results.” (page 11, line 299-303)

Point 8 : Future studies - [line 320+] I think this should be the most important part of the paper. If the authors believe that more work should be done with EA, it would be helpful to make a strong case for such study. The null finding does not seem encouraging, so a more detailed discussion of why one should spend research funds on continuing the work requires a more detailed recommendation. Among the things that might be helpful to someone considering future work:

- Should another pilot study be done, or should a large program be done. If a pilot program is recommended, what is the minimum number of cohorts that should be considered?
- What frequency and duration of treatments should be adopted?
- How long should the study last?
- Should there be a balanced male and female participation?
- Should the study include an EA-only group?

Response 9: Thank you for your valuable comments. In accordance with your suggestions, we have revised the Conclusion section as follows: “Nevertheless, we believe that the results of our study could have varied greatly depending on sample size, frequency and total number of sessions, intervention that is combined with EA, and acupoint specificity. We hope that well-designed RCTs with enough sample size aimed at investigating possible effects or add-on effects of EA on MCI will be conducted in the future.” (Page12, line 334-337)

Reviewer 2 Report

The Authors conducted a  randomized controlled clinical trial investigated the effects of electroacupuncture combined with computer-based cognitive rehabilitation mild cognitive impairment (MCI), showing beneficial effects on improving cognitive function in patients with MCI. 

I have some concerns about the diagnosis of MCI. Why are there any daily living activity scales that are normal in MCIs?

Author Response

We are very grateful for the constructive comments and the opportunity to revise our manuscript.

Point-by-point responses to the reviewers’ comments are provided below.

The Authors conducted a  randomized controlled clinical trial investigated the effects of electroacupuncture combined with computer-based cognitive rehabilitation mild cognitive impairment (MCI), showing beneficial effects on improving cognitive function in patients with MCI

Point 1: I have some concerns about the diagnosis of MCI. Why are there any daily living activity scales that are normal in MCIs?

Response 1: Thank you for your valuable comments. Advanced age involves structural and functional deterioration of most physiological systems which may negatively impact an individual’s ability to carry out activities of daily living such as grooming, feeding, mobilizing, and continence, alongside instrumental activities of daily living such as housework, managing money, and shopping for groceries [2]. Elderly people with MCI may have a negative impact on their activity of daily living due to other causes such as general weakness or depressive mood than cognitive impairment. Thus, we investigate the unexpected effects of EA-CCR on activities of daily living and instrumental activities of daily living.

Reference

  1. Deng, M.; Wang, X.F. Acupuncture for amnestic mild cognitive impairment: a meta-analysis of randomized controlled trials. Acupunct Med 2016, 34, 342–348.
  2. Roberts, C.E.; Phillips, L.H.; Cooper, C.L.; Gray, S.; Allan, J.L. Effect of different types of physical activity on activities of daily living in older adults: systematic review and meta-analysis. J Aging Phys Act 2017, 25(4), 653-670.
